# Oncolytic Activity of Sindbis Virus with the Help of GM-CSF in Hepatocellular Carcinoma

**DOI:** 10.3390/ijms25137195

**Published:** 2024-06-29

**Authors:** Xiangwei Shi, Kangyixin Sun, Li Li, Jingwen Xian, Ping Wang, Fan Jia, Fuqiang Xu

**Affiliations:** 1NMPA Key Laboratory for Research and Evaluation of Viral Vector Technology in Cell and Gene Therapy Medicinal Products, The Brain Cognition and Brain Disease Institute, Shenzhen Institute of Advanced Technology, Chinese Academy of Sciences, Shenzhen 518055, China; 2Guangdong Provincial Medical Products Administration, Shenzhen Institute of Advanced Technology, Chinese Academy of Sciences, Shenzhen 518055, China; 3Shenzhen Key Laboratory of Viral Vectors for Biomedicine, Shenzhen-Hong Kong Institute of Brain Science, Shenzhen Institute of Advanced Technology, Chinese Academy of Sciences, Shenzhen 518055, China; 4Key Laboratory of Quality Control Technology for Virus-Based Therapeutics, Shenzhen Institute of Advanced Technology, Chinese Academy of Sciences, Shenzhen 518055, China; 5Innovation Academy for Precision Measurement Science and Technology, Chinese Academy of Sciences, Wuhan 430071, China; 6University of Chinese Academy of Sciences, Beijing 100049, China; 7Wuhan National Laboratory for Optoelectronics, Huazhong University of Science and Technology, Wuhan 430074, China; 8Shenzhen Institute for Drug Control, Shenzhen 518057, China; 9Center for Excellence in Brain Science and Intelligence Technology, Chinese Academy of Sciences, Shanghai 200031, China

**Keywords:** oncolytic virus, hepatocellular carcinomas, Sindbis virus, tumor microenvironment

## Abstract

Hepatocellular carcinoma is a refractory tumor with poor prognosis and high mortality. Many oncolytic viruses are currently being investigated for the treatment of hepatocellular carcinoma. Based on previous studies, we constructed a recombinant GM-CSF-carrying Sindbis virus, named SINV-GM-CSF, which contains a mutation (G to S) at amino acid 285 in the nsp1 protein of the viral vector. The potential of this mutated vector for liver cancer therapy was verified at the cellular level and in vivo, respectively, and the changes in the tumor microenvironment after treatment were also described. The results showed that the Sindbis virus could effectively infect hepatocellular carcinoma cell lines and induce cell death. Furthermore, the addition of GM-CSF enhanced the tumor-killing effect of the Sindbis virus and increased the number of immune cells in the intra-tumor microenvironment during the treatment. In particular, SINV-GM-CSF was able to efficiently kill tumors in a mouse tumor model of hepatocellular carcinoma by regulating the elevation of M1-type macrophages (which have a tumor-resistant ability) and the decrease in M2-type macrophages (which have a tumor-promoting capacity). Overall, SINV-GM-CSF is an attractive vector platform with clinical potential for use as a safe and effective oncolytic virus.

## 1. Introduction

Hepatocellular carcinomas (HCCs) are the most common type of primary liver cancer and, together with other liver cancers, contribute to the high mortality rate of patients [1]. The principal modalities of cancer treatment encompass surgical intervention, radiation therapy, and chemotherapy, which exert a certain degree of inhibitory effects on tumor growth. However, the emergence of drug resistance, the toxic and side effects of traditional tumor therapy, and the unsatisfactory prognosis in the treatment process have prompted the exploration of new strategies to improve cancer treatments. 

Oncolytic viruses (OVs) can selectively target tumor cells without damaging normal tissues and cells [2,3,4], which makes oncolytic virus therapy a potential cancer therapeutic approach. In recent years, an increasing number of oncolytic viruses have been developed for preclinical and clinical research, such as adenoviruses, poxviruses, herpes simplex viruses, measles viruses, and vesicular stomatitis viruses [5,6,7,8,9,10,11,12,13,14,15,16], offering various advantages in the field of cancer therapy. First, oncolytic viruses can replicate in tumor cells, leading to the death or apoptosis of tumor cells. They can also induce the generation of specific immune responses at the tumor site and throughout the body after the tumor cells are infected [17,18]. Moreover, as a vector, oncolytic viruses can carry cofactors, such as granulocyte-macrophage colony-stimulating factor (GM-CSF), interleukin 12 (IL-12), or some immune checkpoint inhibitor antibodies such as programmed cell death protein 1 (PD-1) and cytotoxic T-lymphocyte associated protein 4 (CTLA4) [3,19,20,21,22,23,24,25,26], and deliver them to tumor lesions for tumor therapy. During this process, immune cells such as mononuclear phagocytes, dendritic cells, CD8+T cells, and Natural killer cells (NK cells) are released and transform an immunologically “cold” tumor microenvironment with few immune cells into a “hot” tumor microenvironment infiltrated by immune cells and cytokines [27,28,29].

The Sindbis virus (SINV) is a member of the Alphavirus genus (the *Togaviridae* family), which is a blood-borne virus that is usually transmitted to mammals through mosquito bites [30]. The SINV genome is approximately 11.7 kb in length and encodes four nonstructural proteins (nsP1-4) and five structural proteins (C, E3, E2, 6K, and E1) [30]. Among them, the nonstructural proteins are produced by cleavage of two polyprotein precursors (P123 and P1234), which together form the replicase and the transcriptase system for viral RNA synthesis [30,31,32]. In previous work, the 67KDa high-affinity laminin receptor (LAMR) protein was shown to bind to the Sindbis virus and was proven to be overexpressed on the surface of many human cancer cells, including liver-derived tumor cells [33]. Recently, the very-low-density lipoprotein receptor (VLDLR), apolipoprotein E receptor 2 (ApoER2), and the CD147 protein complex, which are associated with tumor proliferation and the inhibition of apoptosis, have also been revealed to be associated with the entry of the SINV into cells [34,35]. As a result, the SINV can infect a wide range of mammalian cells, exemplifying the benefits of using it as an oncolytic virus. The SINV, as a positive single-stranded RNA virus, can avoid the risk of insertion of chromosomal mutagenesis and can also be systematically delivered in vivo, remaining highly effective with repeated administration [36,37,38,39,40]. In addition, epidemiologic statistics show that seroprevalence in SINV-endemic areas ranges from 2.9% to 39% [30], while 45% to 98% of the world’s population is seropositive for HSV1 [41] and the symptoms of SINV infection are mild and usually only include fever and joint pain [30]. At this stage, many oncolytic viruses have been well developed, but each of the different viruses has its drawbacks, such as the poor efficacy of HSV in patients with pre-existing neutralizing antibodies and the broad organophilicity of adenoviruses [42]. Therefore, with the development of oncolytic therapies, more viruses need to be explored to accommodate different tumor types and provide personalized treatment options for individual patients.

It is reported that the Sindbis virus has been used in the treatment of cervical cancer, ovarian cancer, and brain tumors with promising results [43,44,45,46,47]. In particular, the SINV has been used as a therapeutic agent in a mouse tumor model constructed from ML-14a (mouse hepatocellular carcinoma cell line) cells [48], and studies have been conducted to target the liver by inserting microRNA response elements (MREs) into the Sindbis virus genome to improve the safety of its use as an oncolytic virus [49]. However, some reports have also pointed to the genetic instability of the SINV, with SINV-containing reporter genes showing the loss of expression of transgenes in offspring viruses [49,50]. To address this problem, we obtained a mutant virus, developed in our previous work, in which the 285th amino acid in an nsP1 was included from a glycine to a serine [50]. Compared with the wild-type progeny virus, which had a 57.3% loss of reporter genes in the generation of P5, the mutation allowed the stable expression of the exogenous gene throughout the P5 generation [50]. Therefore, in this work, we utilized a mutated Sindbis virus vector to express GM-CSF (SINV-GM-CSF) and found that it can effectively slow down the course of liver-tumor-bearing mice and improve the intratumor microenvironment. Overall, this GM-CSF-carrying Sindbis virus mutant is a potential therapeutic tool for liver cancer treatment.

## 2. Results

### 2.1. Sindbis Virus (SINV) Can Selectively Infect and Kill Hepatocellular Carcinoma Cells In Vitro

To evaluate the feasibility of the SINV for Hepatocellular carcinomas (HCCs) treatment, we tested the susceptibility of various hepatocellular carcinoma cells and normal cell lines to the SINV. SINV-EGFP was used to infect BHK-21, Hep3B, Huh-7, HepG2, LX-2, and C2C12 cells, and the results showed that all hepatocellular carcinoma cells (Hep3B, Huh-7, and HepG2) could be infected by SINV-EGFP by observing the expression of green fluorescent proteins (Figure 1A). Obvious pathological effects were observed in these cells, but not in normal human hepatic astrocyte LX-2 cells or mouse myoblast C2C12 cells. These results indicate that SINV-EGFP can selectively infect hepatocellular carcinoma cells, which is favorable for the use of the SINV in liver cancer therapy. Then, we used an MTT cell proliferation and cytotoxicity assay kit to detect cell viability at 48 h post-SINV infection. As shown in Figure 1B, SINV-EGFP infection significantly decreased the cell viability of BHK-21, Hep3B, HuH-7, and HepG2 cell lines, while there was no significant effect on the activity of normal LX-2 and C2C12 cells. There was a significant difference in the infection efficiency of the SINV in three different hepatocellular carcinoma cell lines, and to examine whether these differences were determined by the expression of viral receptors on the cell surface, the expression of viral receptor proteins was examined in these six cell lines. From the results of a Western blot, it can be seen that the expression of two SINV receptors, high-affinity laminin receptor (LAMR) and low-density lipoprotein receptor (LDLR), was not proportional to the efficiency of infection in the six cell lines. 

### 2.2. SINV-GM-CSF Can Effectively Replicate and Express the GM-CSF Protein in Tumor Cells

To enhance the oncolytic effect of the SINV, we inserted granulocyte-macrophage colony-stimulating factor (GM-CSF) into the SINV vector and evaluated the expression of the GM-CSF protein on BHK-21 cells and Hep3B cells in vitro. Both cells were infected with wide-type SINV (SINV-WT) and SINV-GM-CSF at MOIs of 0.1 and 1, respectively. As shown in Figure 2A, the insertion of the GM-CSF gene into the viral vector does not impact viral growth or replication. Moreover, both SINV-WT and SINV-GM-CSF can replicate and produce large numbers of progeny viruses in Hep3B cells. These findings indicate that SINV-GM-CSF can effectively infect tumor cells and sustain the ability to replicate within them. To confirm GM-CSF protein expression after the virus injection, indirect immunofluorescence staining was performed. The anti-GM-CSF antibodies were incubated on BHK-21 and Hep3B cells infected with SINV-GM-CSF. The red fluorescence signals indicate the expression of GM-CSF (Figure 2B), and the results of a Western blot further confirmed the expression of GM-CSF proteins in both cells (Figure 2C). Furthermore, we validated the tumor-killing effect of SINV-GM-CSF in a tumor-bearing mouse model.

### 2.3. SINV-GM-CSF Can Effectively Inhibit the Growth of Tumors in Hep3B Subcutaneous Xenograft Mouse Models 

To establish a more convenient method of detecting tumors in vivo, we constructed Hep3B cell lines expressing luciferase. To study the therapeutic effect of SINV-GM-CSF treatment in vivo, Hep3B-Luc cells were subcutaneously implanted in Nu/Nu nude female mice. After that, mice with similar tumor sizes were randomly divided into three groups and given intratumoral injections of 5 × 10^6^ PFU of SINV, 5 × 10^6^ PFU of SINV-GM-CSF, and 100 uL of PBS (mock-treated control) three times every other day. After 15 days, IVIS living imaging was performed to assess the luciferase signals of tumor cells (Figure 3A). The results showed that the luciferase signals in the treated group were significantly lower than those in the PBS control group, suggesting that while SINV and SINV-GM-CSF both inhibited tumor growth, SINV-GM-CSF was more effective. Furthermore, the fluorescence signal intensity was noticeably reduced in the SINV-GM-CSF group compared to the wild-type group (Figure 3B,C). According to the results of the H&E staining, tumor cells exhibited vigorous growth and were densely arranged, with no apparent tumor cell necrosis in the PBS group. In contrast, in the SINV-WT and SINV-GM-CSF treatment groups, there were larger areas of necrotic tissue observed within the tumor, where the nuclei of cells had dissolved and disappeared (Figure 3D). Together, these data revealed that SINV-WT and SINV-GM-CSF had tumor-killing abilities in Hep3B cell tumor models, suggesting that SINV vectors have great potential as innovative therapeutic agents for treating human hepatocellular carcinomas. 

### 2.4. SINV-GM-CSF Improved the Microenvironment in Hep3B Tumor-Bearing Models

To further study the effect of SINV-GM-CSF in tumor microenvironments during the therapy, single-cell RNA sequencing was performed on tumor tissues. Cell suspensions of tumor tissues from the SINV-GM-CSF group and the control group were obtained after treatment (*n* = 3), and single cells in the two groups were sequenced. The number of filtered cells obtained from cell suspension samples was 13,272 in the PBS control group and 16,246 in the SINV-GM-CSF experimental group. It can be seen from the cell clustering results that hepatocyte cells, endothelial cells, and a variety of immune cells can be detected in both the treatment and control groups, and the number of monocytes and neutrophils significantly increased in the experimental group (Figure 4A). The total number of immune cells in the tumor was further compared between the two groups, and it was found that the proportion of immune cells in the total cells in tumor tissue significantly increased in the SINV-GM-CSF group compared with the control group (Figure 4B); in addition, the number of Natural killer cells (NK cells), Dendritic cells (DCs), B cells, and neutrophils rose (Figure 4C). This result confirmed that SINV-GM-CSF increased the number of DCs and stimulated the production of T cells and B cells after therapy, indicating that SINV-GM-CSF can improve the immune response at the tumor site. It is noteworthy that there was a tendency for the number of macrophages within the tumor to decrease after the SINV-GM-CSF injection. Based on this result, we further investigated the changes in the different types of macrophages in the treatment group. 

In the treatment and control groups for the tumor tissue samples, macrophages were further subjected to subclass-based cluster analyses. CD68-positive cells were classified as macrophages, while cells with double CD68- and CD163-positive cells were classified as M2-type macrophages, and the rest were classified as M1-type macrophages (Figure 4D,E,F). By comparing the two types of macrophages, M1 and M2, in the tumor tissues of the two groups, it was found that the proportion of M1-type macrophages in the tumor of the SINV-GM-CSF experimental group increased. In contrast, the number of M2-type macrophages decreased after the injection (Figure 4G). Previous studies have indicated that the two types of macrophages have opposed roles in tumors: M1-type macrophages have anti-tumor effects, while M2-type macrophages potentially promote tumor growth [51,52], stimulate angiogenesis, and enhance the invasion of tumor cells. Thus, SINV-GM-CSF promoted the anti-tumor effect of oncolytic viruses by inducing an increase in M1-type macrophages and a decrease in M2-type macrophages.

## 3. Discussion

Oncolytic viruses continue to evolve rapidly as highly promising cancer treatment strategies, and Alphaviruses have also been extensively studied due to their natural tumorigenic ability. For example, recombinant M1-based oncolytic viruses, used in the treatment of primary liver cancer, have been designated by the FDA as orphan drugs. In this study, we chose the Sindbis virus, also a member of the Alphavirus genus, to evaluate the therapeutic effect of GM-CSF-carrying sindbis virus (SINV). 

From the results of the virus infecting different cell lines, it can be observed that only a few LX-2 and C2C12 cells expressed fluorescence, and no cytopathic effect was observed in these cells compared to the three hepatocellular carcinoma cell lines. In addition, the results of the Western blot indicated that these two cells can provide receptors for the virus to enter the cells, which may be attributed to the fact that in normal cells, the presence of the SINV led to the activation of the interferon type I pathway, resulting in the activation and release of Protein kinase R (PKR), which inhibited the translation and replication of the viral genome [53,54]. Previous work found that the SINV is more sensitive to PKR-negative mice, and this provides an advantage for the SINV in the treatment of some PKR-deficient tumors. [32,55,56]. In addition, regarding matrix remodeling-associated 8 (MXRA8), LAMR has been identified as a cell entry receptor for the SINV [57,58], while VLDLR and ApoER2 are also recognized as viral receptors for the SINV [35], and when all of these receptors are lowly expressed, the SINV can also use the natural resistance-associated macrophage protein (NRAMP) as an alternative receptor [59]. The receptor diversity of the SINV determines its broad host range, which explains why the receptor expression profiles of LDLR and LAMR from several cell types could not be correlated with the efficiency of viral infection (Figure 1D).

In this work, we combined viruses and cytokines to construct SINV-GM-CSF and analyzed its effect on the tumor microenvironment and found that the SINV-GM-CSF improved the tumor microenvironment during treatment. After viral infection, a strong antiviral response at the tumor site translates into active inflammation in the tumor, which indirectly triggers an anti-tumor response, thereby causing the activation of innate immune cells, including dendritic cells and NK cells [60]. In this process, the SINV enhances CD8+ T cells and NK cells in the immune system, which has been previously reported [61,62], and GM-CSF enhances the presentation of antigens by activating DCs, thereby stimulating the immune response and increasing the number of T cells, B cells, and NK cells [63,64,65], which is consistent with the results of single-cell sequencing (Figure 4A). Moreover, M1-type macrophages can reverse the immunosuppression of the tumor microenvironment and restore the activity of CD8+ cytotoxic T cells [51,52,66,67,68]. Monocytes further differentiate into macrophages and dendritic cells to generate immune responses, and increasing the number of intratumoral monocytes may further increase the number of macrophages and dendritic cells in the tumor. This process leads to an increase in myeloid lineage cells after treatment, which is not only due to the release of GM-CSF from SINV-GM-CSF, but the virus itself is also involved in this process. In our results, the number of macrophages was similar in the treatment and control groups, which may have been caused by a higher number of M2-type macrophages in the PBS group than in the treatment group. In addition, in the early stages of viral infection, macrophages exert antiviral functions by producing interferon type I, limiting viral spread and exertion [69,70,71], and phagocytosing infected tumor cells [72], which may lead to local depletion of macrophages, resulting in a lower macrophage population after viral infection. 

In addition, as shown in Figure 4G, compared with the PBS group, the percentage of M1-type macrophages was elevated, and the M2 type was decreased in the treatment group. We assumed that this may be due to the presence of oncolytic viruses during therapy allowing macrophages to achieve a phenotypic shift from a tumor-supportive M2 to a pro-inflammatory M1 phenotype, as previously reported [73]. In virotherapy, the role of macrophages is complex, as macrophages can either mediate antiviral immunity to generate obstructive macrophage responses, which block viral transmission, or recruit more immune cells to infiltrate the tumor and help improve the local tumor microenvironment. The balance between antiviral and anti-tumor responses determines the final therapeutic outcome. Based on our results, considering the percentage of M1- and M2-type macrophages in total macrophages, SINV-GM-CSF is suggested to enhance the tumor-killing ability by modulating the intratumor microenvironment.

In the course of tumor therapy, due to the heterogeneity of the tumor, the use of single virus therapy for solid tumor treatment may have limitations, which makes it necessary to constantly explore more research methods to achieve the conquest of cancer. As an oncolytic virus, the loading capacity of the SINV allows SINV vectors to enhance oncolytic effects by combining various immunotherapies. In addition, the potential application of SINV-GM-CSF in treating various cancers is promising for future studies, as the SINV can reach the entire body by intraperitoneal or intravenous injection, and this hematogenous characteristic makes it advantageous for the treatment of metastases and some microscopic lesions. 

## 4. Materials and Methods

### 4.1. Plasmid Construction

The plasmids of pSINV, pSINV-EGFP, and pSINV-GM-CSF were constructed in our previous research [50]. Those plasmids have the same mutation in the nsP1 gene with G285S, and this vector facilitates the stable expression of exogenous genes [50]. The EGFP (Genbank: OQ870305) and GM-CSF (Genbank: NM_009969) genes were inserted between ApaI and NotI under the control of a second sub-promoter. All plasmids have been verified by DNA sequencing.

### 4.2. Cells and Viruses

All experiments regarding the SINV were performed in a Biosafety Level 2 laboratory. The cell line we used to amplify the virus was baby-hamster kidney cells (BHK-21, American Type Culture Collection, ATCC, USA). The Hep3B, LX-2, C2C12, HuH-7, and HepG2 cells were purchased from Procell Life Science & Technology (Wuhan, China). These cells have STR identification certificates to avoid misidentification. In addition, all cells were tested with the MycAwayTM Plus-Color One-Step Mycoplasma Detection Kit (#40612ES25, Yeasen, Shanghai, China) with negative results.

BHK-21, LX-2, C2C12, and HuH-7 were cultured in Dulbecco’s Modified Eagle’s Medium (#11965092, Thermo Fisher, Waltham, MA, USA) containing 10% fetal bovine serum (#10099158, FBS, Thermo Fisher, Waltham, MA, USA) and 1% penicillin-streptomycin (#15140122, P/S, Thermo Fisher, Waltham, MA, USA); Hep3B and HepG2 cells were cultured in Minimum Essential Medium (#11095080, ThermoFisher Thermo Fisher, Waltham, MA, USA) with 10% FBS and 1% P/S. The Hep3B-Luc cell line was constructed by a lentiviral vector that expressed firefly luciferase and EGFP. After infection of Hep3B cells and flow cytometry, we obtained Hep3B-Luc cells that stably expressed luciferase.

The pSINV-WT, pSINV-EGFP, and pSINV-GM-CSF were transfected into the BHK-21 cells with Lipofectamine 2000 reagent (#11668030, Thermo Fisher Thermo Fisher, Waltham, MA, USA). After 6 h, the supernatant was replaced by DMEM containing 2% FBS at 37 °C in 5% CO_2_. The viruses were collected from the supernatant 48 h post-transfection. Virus titers were measured by plaque assay and counted as plaque-forming units (PFU).

### 4.3. Cell Viability Assay

Cell viability was detected by using an MTT cell proliferation and cytotoxicity assay kit (#C0009S, Beyotime, Nanjing, China). The cells were inoculated in 96-well plates in advance and infected with SINV-GM-CSF at 1 multiplicity of infection (MOI), while the control group was infected with PBS (n = 4). After being infected for 48 h, 10 μL of MTT solution (5 mg/mL) was added to each well and incubated in the cell incubator for 4 h. Then, 100 μL of a formazan solution was added and mixed, incubating for 3-4 h until the formazan was completely dissolved. The absorbance near 570 nm was then measured with an enzyme-labeled instrument, with lower values indicating greater cytotoxicity.

### 4.4. Western Blot Analysis

Cell samples were collected followed by the addition of 1×SDS-PAGE loading buffer, after which samples were placed in a metal bath at 98 °C for 5 min, and proteins were loaded into the 10% SDS-PAGE gels for separation. The proteins were then transferred to PVDF membranes. The primary antibodies were then incubated overnight at 4 °C, and the secondary antibodies were then probed for 2 h at room temperature. The following primary antibodies were used in this experiment: GM-CSF polyclonal antibody (#ab300495, Abcam, Cambridge, UK), LDLR antibody (#66414-1-Ig, Proteintech, Manchester, UK), LAMR1 polyclonal antibody (#67324-1-Ig, Proteintech, Manchester, UK), and GAPDH monoclonal antibody (#60004-1-Ig, Proteintech, Manchester, UK). Secondary antibodies include HRP-conjugated Affinipure Goat Anti-Rabbit IgG and HRP-conjugated Affinipure Goat Anti-Mouse IgG. Image J was used for the quantification of image bands.

### 4.5. Animal Models and Treatment Methods

All experimental procedures were approved by the Institutional Animal Care and Use Committees at the Shenzhen Institute of Advanced Technology, the Chinese Academy of Sciences. Nu/Nu nude female mice (3–4 weeks old) used in this study were from Hunan SJA Laboratory Animal Company (Changsha, China). For the establishment of tumor models, 4 × 10^6^ Hep3B (*n* = 3 for each group) or Hep3B-Luc cells (*n* = 5 for each group) with 100 μL PBS were subcutaneously injected into the inguinal region of the nude mice. After 7 days, the mice were divided into three groups randomly and injected intratumorally with SINV-WT (5.0 × 10^6^ PFU per 100 μL), SINV-GM-CSF (5.0 × 10^6^ PFU per 100 μL), and PBS (100 μL), respectively, three times every other day. During the treatment process, the physical condition of the experimental animals and tumor growth were continuously observed, and mice with rapid weight changes or excessive tumor size were euthanized. The volume of tumors was measured with a caliper every day using the following formula: 1/2 × length (mm) × width (mm) × width (mm).

### 4.6. IVIS Imaging 

To monitor the process of tumor development in living mice, we utilized the Small Animal In Vivo Imager System (IVIS) to observe the luciferase expression in tumor cells. A concentration of 150 mg/kg (luciferin/body weight) of D-luciferin potassium salt (#40902ES02, Yeasen, Shanghai, China) was administered intraperitoneally to mice before imaging. The quantification of luciferase expression was performed using Living Image version 4.2.

### 4.7. Single-Cell RNA Sequencing

After taking the tumor tissue from each group, the pre-cooled saline was used to wash the tumor tissue quickly to remove the blood cells. After tearing the tumor tissue into small pieces, collagenase and trypsin were used to digest the tumor tissue at 37 °C for 20–30 min. The enzyme digestion was terminated with RPMI 1640 medium, and the single-cell suspension was prepared by centrifugation and resuspension with pre-cooled PBS. Next, we mixed three individual mouse tumor cell suspension samples from each group and performed them as a whole in the single-cell sequencing process. Single-cell capture and library preparation were completed by the BD RhapsodyTM single-cell analysis system (BD Biosciences, Franklin, NJ, USA). The single-cell suspension was loaded into the BD Rhapsody cartridge, and single-cell mRNA capture was achieved by magnetic beads with 200,000 micropores and a barcode for capturing oligonucleotides. Then, the magnetic beads were collected according to the BD RhapsodyTM whole transcriptome analysis (WTA) amplification kit process for cDNA synthesis and library construction. Finally, the library was quantified using the Agilent 2100 Bioanalyzer system (Agilent, Santa Clara, CA, USA) and Qubit 4.0 Fluorometer (Thermo Fisher Scientific, Waltham, MA, USA) and sequenced on the Illumina NovaSeq 6000 (Illumina, San Diego, CA, USA) with 300 bp reads (150 bp paired-end reads). 

The data analysis of single-cell RNA sequencing involves multiple steps, including quality control, alignment, and clustering. Quality control was conducted on the data to extract the barcode and UMI, which were then compared with the mouse reference genome (GRCm38-PhiX-gencodevM19). The resulting matrix containing cell index and gene expression level information was imported into Seurat (v.4.0.3) for subsequent analysis. The screening criteria were defined as follows: nGene > 1000, nUMI > 1000, log10GenesPerUMI > 0.80, mitoRatio < 0.25; DoubletFinder was utilized to eliminate double cells. The filtered cell expressions were normalized, and differentially expressed genes (DEGs) were selected for principal component analysis (PCA) dimensionality reduction; t-distribution random neighbor embedding (tSNE) was employed to visualize the clustering outcomes. Marker genes were identified using the default parameters of Find All Markers in Seurat. The original clusters were annotated using the Mouse RNA seq Data dataset in Single R (v1.0.1). Based on these annotated clusters, subset functions were applied to extract macrophage clusters for further subanalysis purposes. To enhance cell-type discrimination, the ImmGenData dataset was used for cluster annotation.

### 4.8. Statistical Analysis

The data were presented as mean ± S.E.M., and GraphPad Prism 8.0 was utilized for the processing of all graphs and statistical analysis.

## 5. Conclusions

In this study, we constructed an SINV vector carrying the GM-CSF gene and applied it to hepatocellular carcinomas. Most of the previous studies on the SINV for hepatocellular carcinoma have focused on the cellular level without further follow-up in vivo experiments or mechanistic studies [48,74]. Our work comprehensively evaluates the effect of the SINV in liver cancer treatment based on this foundation, from in vivo, ex vivo, and the tumor microenvironment. At the vector level, we introduced a new vector of the Sindbis virus carrying GM-CSF and found that both the SINV and SINV-GM-CSF were effective in killing hepatocellular carcinoma cells, and SINV-GM-CSF stimulated the immune response of the tumor microenvironment by increasing the number of T cells and NK cells as well as enhancing the M1 macrophage and DCs in the process of treatment, thus further enhancing the anti-tumor effect. These findings not only demonstrate the potential of the SINV-GM-CSF mutant as a replicable oncolytic virus, but also provide a basis for future studies of the SINV as well as other Alphaviruses for the treatment of hepatocellular carcinoma.

## Figures and Tables

**Figure 1 ijms-25-07195-f001:**
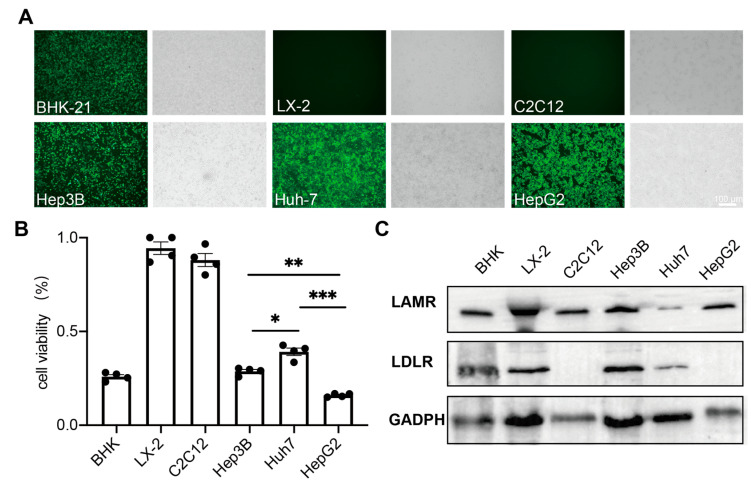
SINV can selectively infect and kill hepatocellular carcinoma cells in vitro. (**A**) Hepatocellular carcinoma cells and normal tissue cells were infected with SINV-EGFP (MOI = 1), and representative images were obtained at 24 h post-infection, where BHK-21 cells were used as a positive control. Scale bar 100 μm. (**B**) All cells were infected with SINV-EGFP, and cell viability was assessed at 48 h post-infection by the MTT cell proliferation and cytotoxicity assay kit. The data are expressed as the mean ± SEM of four independent experiments (* *p*  <  0.05, ** *p*  <  0.01, *** *p*  <  0.001, *n* = 4). (**C**) The result of LAMR and LDLR expression in 6 cell lines, where the expression of GADPH was used as a reference.

**Figure 2 ijms-25-07195-f002:**
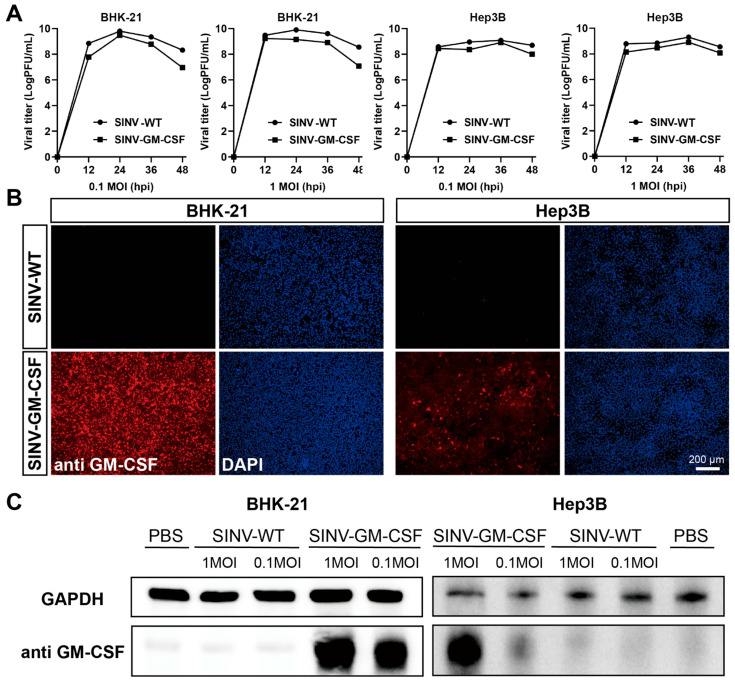
SINV-GM-CSF can effectively replicate and express the GM-CSF protein in tumor cells. (**A**) The one-step growth curves of SINV-WT and SINV-GM-CSF in BHK-21 cells and Hep3B cells, respectively. The viruses were collected and titered on BHK-21 cells at the indicated time points. GraphPad Prism 8.0 was used for statistical graphs. (**B**) The viruses were infected with BHK-21 cells and Hep3B cells, and the red fluorescence indicated GM-CSF protein expression by anti-GM-CSF. DAPI was used to stain the nucleus. Scale bar 200 μm. (**C**) Western blot analysis was performed for GM-CSF protein expression after treatment with 0.1 and 1MOI of SINV-WT and SINV–GM–CSF, respectively. GAPDH was used as a control for protein loading. One representative image of three experiments is shown.

**Figure 3 ijms-25-07195-f003:**
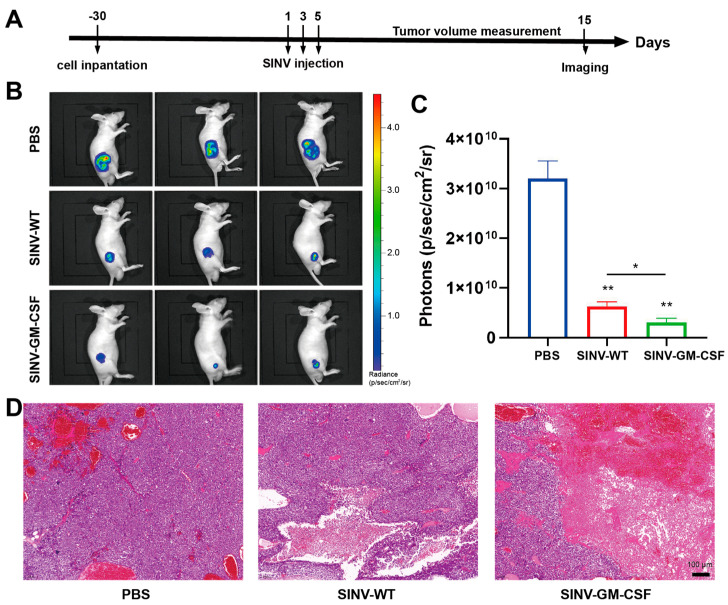
SINV-GM-CSF can effectively inhibit the growth of tumors in Hep3B subcutaneous xenograft mice models. (**A**) Schematic of SINV-GM-CSF treatment in hepatocellular carcinoma tumor models. (**B**) Luciferase imaging of tumors 15 days after SINV treatment (*n* = 3). (**C**) The quantitative result of luciferase in the (**B**) plot. Data are expressed as mean ± SEM, *n* = 3 mice per group, two-tailed unpaired t-test with Welch correction (* *p*  <  0.05, ** *p*  <  0.01, *n* = 3). (**D**) H&E staining of the PBS control group and the SINV-WT, SINV-GM-CSF-treated groups. Scale bar 100 μm.

**Figure 4 ijms-25-07195-f004:**
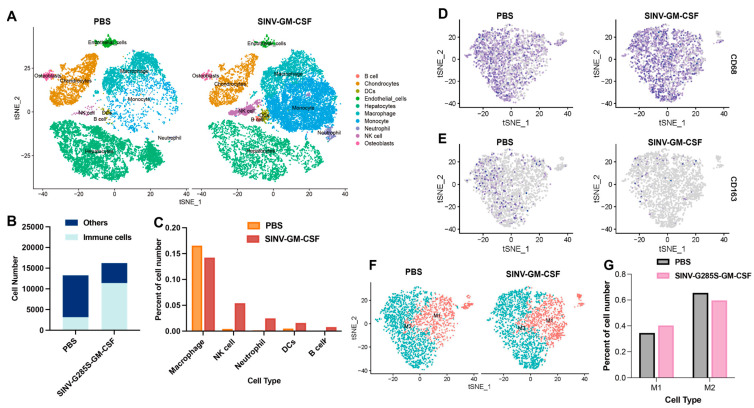
SINV-GM-CSF improved the immune microenvironment and induced an increase in M1-type macrophages. (**A**) Cell clustering of tumor tissue in the SINV-GM-CSF treatment group and PBS control group. (**B**) The proportion of immune cells in all cells in the tumor tissue. (**C**) Comparison of the number of immune cells of different subtypes in the two groups of tumor tissues, *n* = 3 per group. (**D**) Distribution of the CD68 marker gene in macrophages in PBS and SINV-GM-CSF groups. (**E**) Distribution of the macrophage CD163 marker gene in two groups of tumor tissues. (**F**) The clustering of M1 and M2 macrophages in two groups of tumor tissues. (**G**) Quantification of the number of M1 and M2 macrophages in two groups of tumor tissues.

## Data Availability

The data that support the findings of this study are available from the corresponding author upon reasonable request.

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
