# Peer review of "Oncolytic Activity of Sindbis Virus with the Help of GM-CSF in Hepatocellular Carcinoma"

_ijms, 2024, doi:10.3390/ijms25137195_

Round 1

Reviewer 1 Report

Comments and Suggestions for Authors

Comments on the Quality of English Language

Extensive editing of English language required.

Author Response

Dear reviewer,

Thank you for your constructive comments on the manuscript. We have carefully considered the suggestion and make some changes in resubmitted manuscript.

Please see the attachment for detailed responses to the comments.

Reviewer 2 Report

Comments and Suggestions for Authors

The manuscript describes a model of oncolytic virus (Sindbis virus) as an immunotherapy approach to treat hepatocellular carcinoma cancer. It is well written, with diverse experimental approaches and conclusions corresponding to the data obtained.

Minor comments:

1 Introduction

It is not very clear if SINV is more likely to infect tumor cells than other healthy ones. As told in the discussion there are several receptors for the virus, and they are expressed in various tissues. Possible advantages of this virus against others models more frequently employed (adenovirus, etc…) should be also commented.

2 Material and methods

79  “…The nsP1 gene of pSINV-EGFP and pSINV-GM-CSF were mutated with [41].”

Not clear to me, ¿Something missing?

2 Results

Figure 1. Details of virus mutants (position of mutation, genes, etc…) are not detailed in the present manuscript, so the meaning of genes in panel A are not known. It should be clarified in the text y/o figure. Eve better if some basic information about virus model is added in the introduction.

A common critique to the experimental approach shown in Figures 4 and Figure 5.  I think the right control for the SINV-GM-CSF infection  is not PBS. The “wt” virus should have been included since, as a foreign invader, it is going to stimulate the immune response whatsoever, and be responsible, at least in part, of the alterations in the immune status. The exact influence of GM-CSF cannot be fully elucidated in these experiments.

Figure 4. Bar colors in panels B and C do not stand for the same and are consequently a bit confusing.

In the group infected with the SINV-GM-CSF virus, one would expect potentiation of myeloid lineage cells, but macrophages are even diminished against control (PBS). This apparent contradiction should be discussed.

No statistics included in the bars.

Figure 5. As in figure 4, panels C and D are a bit confusing since color bars in both panels do not represent the same thing. .

It is a bit difficult to really distinguish differences in M1/M2 subtypes between both groups (PBS vs SINV-GM-CSF) since, once more, no statistics are included. Just “by eye”, panel B indicates much less M2 macrophages (CD183 positive staining) in SINV-GM-CSF vs PBS treated cells. In panel D, this difference does not seem to be so significative.

Main text indicates that “Thus, SINV-GM-CSF promoted the anti-tumor effect of oncolytic viruses by inducing an increase in M1-type macrophages and a decrease in M2-type macrophages”, but in any case, the proportion of M2-“suppressor” macrophages is still higher that the M1, and we cannot forget that, as Figure 4 states, the proportion of macrophages are even reduced in the SINV-GM-CSF group. Authors should discuss these results.

4. Discussion

260  “…and no CPE…” Better “ …and no cytopathic effect… “

273  Previous studies have found that viruses are more susceptible to infecting PKR-negative mice, and SINV is no exception to this rule, proving the benefits of SINV as an oncolytic virus [46-48]”

I do not understand authors intend to say in this sentence.

Results of figure 4 and 5 should be discussed, as aforementioned.

A discussion of futures perspectives of Sindbis virus against other models of oncolytic viruses  in hepatocellular carcinoma (and other cancer types), would increase interest to this part.

Author Response

(The authors gave the same response as above.)

Round 2

Reviewer 1 Report

Comments and Suggestions for Authors

The revision has solved my previous concerns, and I think it can be published as it is.

Comments on the Quality of English Language

Minor editing of English language is required.

Author Response

Thank you very much for your suggestions.

This manuscript was edited in language at MDPI before we submitted it. To further improve the quality of the manuscript, we have scrutinized the text and have made minor changes in the resubmitted manuscript to the best of our ability, such as singular and plural changes, the addition of articles, and corrections to the spelling of words. These changes are highlighted in blue in the resubmitted manuscript.

Reviewer 2 Report

Comments and Suggestions for Authors

Author Response

Thanks for your comments.

We have revised the manuscript based on your excellent comments, please see the attachment for details.
